# Selection and Validation of Reference Genes for Gene Expression Studies Using Quantitative Real-Time PCR in Prunus Necrotic Ringspot Virus-Infected *Cucumis sativus*

**DOI:** 10.3390/v14061269

**Published:** 2022-06-10

**Authors:** Zhenfei Dong, Binhui Zhan, Shifang Li

**Affiliations:** 1State Key Laboratory for Biology of Plant Diseases and Insect Pests, Institute of Plant Protection, Chinese Academy of Agricultural Sciences, Beijing 100193, China; dongzhenfeiqaz@163.com; 2Department of Fruit Science, College of Horticulture, China Agricultural University, Beijing 100193, China

**Keywords:** reference genes, real-time PCR, gene expression, *Cucumis sativus*, virus-infected

## Abstract

Several members of the genus *Ilarvirus* infect fruit trees and are distributed worldwide. Prunus necrotic ringspot virus (PNRSV) is one of the most prevalent viruses, causing significant losses. *Cucumis*
*sativus* can be infected by several ilarviruses, leading to obvious symptoms, including PNRSV, which suggests that cucumbers could be good hosts for the study of the pathogenesis of ilarviruses. Real-time quantitative PCR is an optimal choice for studying gene expression because of its simplicity and its fast and high sensitivity, while its accuracy is highly dependent on the stability of the reference genes. In this study, we assessed the stability of eleven reference genes with geNorm, NormFinder, ΔCt method, BestKeeper, and the ranking software, RefFinder. The results indicated that the combined use of *EF1α* and *F-BOX* was the most accurate normalization method. In addition, the host genes *AGO1*, *AGO4*, and *RDR6* were selected to test the reliability of the reference genes. This study provides useful information for gene expression analysis during PNRSV infection and will facilitate gene expression studies associated with ilarvirus infection.

## 1. Introduction

Gene expression analysis is one of the important and efficient approaches to investigating gene function and revealing phenotypic differences in biological research [1]. Microarray analysis [2], transcript profiling [3], and real-time quantitative polymerase chain reaction (RT-qPCR) [4] are common methods used to identify gene expression patterns and compare RNA levels at transcriptional levels among different populations. In particular, RT-qPCR has been widely utilized in the quantification of the expression levels of single or several genes because of its simplicity, speed, sensitivity, accuracy, and relatively low cost [5]. RT-qPCR has also been used to evaluate the accuracy of microarray analysis and transcriptome data [6]. The selection of appropriate reference genes to ensure the accuracy of transcript normalization is both a prerequisite and the most prominent challenge in the performance of RT-qPCR analyses [7]. The use of reference genes that are not stable has a significant impact on the results, even leading to different conclusions [8]. The ideal reference genes are usually stably expressed across all the compared samples or groups, regardless of the experimental conditions, tissue differences, or treatments [9]. However, there is no universal gene that can satisfy all of the requirements [10]. Some housekeeping genes, including *ACTIN*, *TUBULIN*, *UBIQUITIN*, and *GAPDH*, ae the most commonly used reference genes in RT-qPCR tests; however, they vary substantially under different conditions [11,12,13]. A growing body of research indicates that it is critically important to evaluate the stability of the candidate reference genes experimentally for accurate quantification analysis case-by-case, which depends on specific species, the sample or tissue type, different treatments, experimental conditions, etc. [14,15].

Recently, a number of tools have been developed to conduct the comparison of stability analyses of candidate genes and help to select the most suitable reference genes. The free online analytical programs, geNorm (http://medgen.ugent.be/~jvdesomp/genorm, accessed on 8 December 2021) [16], NormFinder (http://www.mdl.dk/publicationsnormfifinder.htm, accessed on 10 December 2021) [17], and BestKeeper (http://www.gene-quantifification.de/bestkeeper.html, accessed on 15 December 2021) [18] are usually used in combination based on different algorithms. Other tools, such as RefFinder (http://blooge.cn/RefFinder/, accessed on 20 December 2021) [19] and RankAggreg [20], can be used to calculate the ranks of candidate genes. These tools have been successfully used to validate reference genes in different plant species, such as *Nicotiana tabacum* [21], *Eleusine indica* [22], *Conyza bonariensis* [23], *Conyza canadensis* [23], *Alopecurus myosuroides* [13], *Sorghum bicolor* [24], *Avena fatua* [25], *Salix matsudana* [26], and *Lilium* spp. [27], under a variety of conditions, including biological and abiotic stress.

Prunus necrotic ringspot virus (PNRSV) is a member of the genus *Ilarvirus* in the family *Bromoviridae*, which was first identified in peach (*Prunus persica* L.) in 1941. It is one of the most economically important and prevalent viruses, and mainly infects stone-fruit trees, including peach [28], nectarine (*P*. persica) [29], almond (*P*. dulcis) [30], sweet cherry (*P*. avium) [31], sour cherry (*P*. cerasus) [32], plum (*P*. domestica) [33], and apricot (*P*. armeniaca) [34], causing significant losses; it can also infect apple (*Malus domestica*) [35], rose (*Rosa chinensis*) [36], cucumber (*Cucumis sativus*) [37], hops (*Humulus*) [38], etc. According to research by Pallas et al., PNRSV infection in *Prunus* species can reduce bud-take in nurseries, decrease fruit growth by 10–30%, induce fruit-yield losses by 20–60%, and delay fruit maturity [39]. Recently, many studies focused on the development of viral detection techniques and the culture of virus-free propagation materials, which reduced the incidence of PNRSV greatly in the production [40]. However, there is little documented evidence as to its pathogenic mechanism. Cucumber is the natural host of PNRSV and shows symptoms of mosaic leaves and severe stunting once infected (Figure 1) [41]. Furthermore, cucumber can also be infected by other ilarviruses from woody plants, which cause obvious symptoms in laboratory conditions involving agrobacterium-mediated infectious clones, such as apple mosaic virus [42] (ApMV), apple necrotic mosaic virus (ApNMV, data not shown), and prune dwarf virus [43] (PDV). Furthermore, the availability of the complete genomic sequence of cucumber and the successful application of virus-induced gene silencing (VIGS) vectors on cucumber, such as cucumber green mottle mosaic virus (CGMMV)-based [44] and tobacco ringspot virus (TRSV)-based vectors [45], has also contributed to the functional genomics research on cucumber and increased the utilization of cucumber as a model plant in the mechanistic research on ilarviruses from woody plants.

In this study, eleven potential reference genes (*GAPDH*, *L23*, *PP2A*, *F-BOX*, *EF1α*, *TUA*, *CYP*, *CACS*, *TIP41*, *UBI-1*, and *UBI-ep*) were selected according to previous research, and we evaluated which of them would be suitable for normalization analysis in specific conditions (at 12 days post-PNRSV-infection) in *C. sativus*. After the validation of the primer specificity, four programs (geNorm, NormFinder, ΔCt, and BestKeeper) and the ranking software RefFinder were used to analyze the stability of the candidate reference genes, which indicated that *EF1α* and *F-BOX* used in combination were the most accurate normalization method. In addition, the three key host genes associated with the viral infection, *AGO1*, *AGO4*, and *RDR6*, were selected to test the reliability of the reference genes. This study is expected to provide a basis for reference-gene normalization in the context of PNRSV infection in *C. sativus* in future gene-expression research.

## 2. Materials and Methods

### 2.1. Plant Materials and Virus Inoculation

Seedlings of *C. sativus* were planted in a growth room under control conditions (28 °C day and 24 °C night, 16 h light and 8 h dark). The viral inoculation was conducted as previously reported. The PNRSV infectious clones with two vectors (one included RNA1 and RNA2 of PNRSV isolate Pch12, and another included RNA3 of PNRSV isolate Pch12) were transformed into *Agrobacterium tumefaciens* EHA105 and grown in Luria-Bertani medium containing kanamycin (100 mg/L) and rifampicin (25 mg/L). The agrobacterium cells were harvested by centrifugation and resuspended in the infiltration buffer (10 mM MgCl_2_, 10 mM MES, and 150 μM acetosyringone) [37]. After incubation for 2 h at room temperature, the culture was diluted to an optical density of 1.0 at OD_600_ nm. After mixing at equal volumes, the cultures were agroinfiltrated into the two expanded cotyledons of seedlings that were about 1 week old. The agroinfiltrated plants were grown in the same conditions.

### 2.2. Total RNA Isolation and First-Strand cDNA Synthesis

Total RNA was extracted from the new leaves of mock-inoculated or PNRSV-inoculated cucumber seedlings 12 days post-inoculation (dpi), using TRIzol reagent (TianGen, Beijing, China), according to the manufacturer’s instructions. One ug of total RNA was treated with DNase I and then a synthesis of first-strand cDNA was performed with random primer and M-MLV reverse transcriptase, using the PrimeScriptTM RT reagent Kit with gDNA Eraser (Perfect Real Time) (RR047A, Takara, Dalian, China). The mixture was incubated at 37 °C for 30 min and 85 °C for 5 s. The resulting cDNA can be stored at −80 °C for long-term use.

### 2.3. Selection of Candidate Reference Genes and Primer Design

Eleven widely used reference genes (*GAPDH*, *L23*, *PP2A*, *F-BOX*, *EF1α*, *TUA*, *CYP*, *CACS*, *TIP41*, *UBI-1*, and *UBI-ep*) were selected for this study. The reference sequences of these genes can be obtained from the NCBI database, and the accession numbers are listed in Table 1. The primer pairs of *GAPDH*, *L23*, *PP2A*, and *F-BOX* were designed using DNAMAN software, version 5.0 (Lynnon Biosoft, Quebec, QC, Canada). The conditions were set with the following parameters: melting temperature of 58–62 °C, GC content of 40–60%, primer length of 20–25 nt, and amplicon length of 80–250 bp. The specific primers of *EF1α*, *TUA*, *CYP*, *CACS*, *TIP41*, *UBI-1*, and *UBI-ep* were based on previous studies. The primer sequences and the associated parameters are listed in Table 1. The specificity of primers was evaluated by PCR amplification using cucumber cDNA as template. The amplified products were subjected to 2% agarose gel electrophoresis and sequenced to verify the targeted gene.

### 2.4. RT-qPCR Analysis

After the validation of primer specificity, RT-qPCR reactions were carried out in 20 µL volume containing 10 µL of TB Green Premix Ex Taq II (Tli RNaseH Plus) (2×), 0.8 µL of each primer (10 µM), 0.4 µL of ROX Reference Dye II (50×), 2 µL of template (the diluted cDNA of different samples), and 6 µL of distilled water. Every treatment contained at least three biological replicates with three technical replicates in clear 96-well or 384-well plates. The RT-qPCR analyses were performed in the Applied Biosystems QuantStudio 6 Flex Real-Time PCR systems (Thermo fisher scientific, Waltham, MA, USA). The PCR program was set as follows: 95 °C for 30 s (denaturation), followed by 40 cycles of 5 s at 95 °C, 20 s at 58 °C, and 35 s at 72 °C, before, finally, dissociation stage was added to generate a melting curve (95 °C for 15 s, 60 °C for 1 min, and 95 °C for 15 s) to verify the specificity of PCR amplification. The E and R^2^ were calculated from the raw data using LinRegPCR software (Academic Medical Centre, Amsterdam, the Netherlands).

### 2.5. Data Analyses for Expression Stability

Four tools, based on different algorithms, were selected to evaluate the stability of the 11 candidate reference genes, as follows: geNorm, NormFinder, ΔCt, and BestKeeper. The Ct values were obtained from the QuantStudio 6 Flex Real-Time PCR systems and exported into an Excel datasheet (Microsoft Excel 2013). The Ct values were directly used or converted into the desired form according to the requirements of the programs. For geNorm and NormFinder analysis, the Ct values were converted into the relative quantities using the formula 2^−ΔCt^ (ΔCt = Ct_sample_ − Ct_mininum_). For BestKeeper analysis, the raw Ct values were directly used. The geNorm software, which is based on the principle that the expression ratio of two reference genes should be constant in all samples regardless of different groups, calculates the gene-expression M-value as the average pairwise variation of one gene with the other candidate genes. The cut-off value of 1.5 is used to assess the stability of the reference genes and the gene with lowest M-value refers is the most stable. In addition, geNorm calculates the optimal number of reference genes required for accurate normalization by pairwise variation (V_n/n+1_). The value of V_n/n+1_ < 0.15 indicates that no extra reference genes are required for normalization. NormFinder is used to rank the stability of the reference genes by the parameters of stability value (SV) within and among groups. Reference genes with the lowest SVs are considered the most stable reference gene. The ΔCt method identifies the potential reference genes by comparing the relative expression between gene pairs. BestKeeper analyses determine the most stable reference genes with the lowest coefficient of variation (CV) and the lowest relative standard deviation (SD). The candidate genes with SD higher than 1.0 are considered inconsistent and are cut off. The r and *p*-value are also important parameters to be considered. RefFinder is an additional web-based tool, which was used to comprehensively rank the order of the 11 candidate genes.

### 2.6. Relative Quantification of AGO1, AGO4, and RDR6

The *EF1α* and *F-BOX*, the two most stably expressed genes, were used in combination as reference genes for the relative quantification of *AGO1*, *AGO4*, and *RDR6*. The geometric mean of the Ct values of two reference genes (multiplying the Ct values of the two reference genes and then taking the square root) was used as the reference Ct value for each biological replicate. The mock-inoculated samples were used as control and the relative quantifications of *AGO1*, *AGO4*, and *RDR6* in PNRSV-inoculated samples were calculated by 2^−ΔΔCt^ method. The experiments were replicated three times, with at least three plants each time.

### 2.7. High-Throughput RNA-Sequencing and Analysis of the Sequence Data

Total RNA was isolated and purified using TRIzol reagent (Invitrogen, Carlsbad, CA, USA) and the RNA amount and purity of each sample were quantified using NanoDrop ND-1000 (NanoDrop, Wilmington, DE, USA). The RNA integrity was assessed by Bioanalyzer 2100 (Agilent, Santa Clara, CA, USA) with RIN number >7.0, and confirmed by electrophoresis with denaturing agarose gel. Poly (A) RNA was purified from 50 μg total RNA with Dynabeads Oligo (dT) 25-61005 (Thermo fisher scientific, Waltham, MA, USA), using two rounds of purification. Finally, we performed the 2 × 150 bp paired-end sequencing (PE150) on an Illumina Novaseq™ 6000 platform (Illumina, Inc., San Diego, CA, USA). Row reads used fastp software (https://github.com/OpenGene/fastp, accessed on 11 June 2021) to remove the reads that contained adaptor contamination, low-quality bases, and undetermined bases with default parameter. The mapping of clean reads onto the *C. sativus* reference genome was conducted using HISAT2 (http://daehwankimlab.github.io/hisat2, accessed on 12 June 2021). Next, StringTie (https://ccb.jhu.edu/software/stringtie, accessed on 12 June 2021) was used to perform expression level for all mRNAs from input libraries by calculating FPKM (total exon fragments /mapped reads (millions) × exon length (kB)). RNA-seq data with details of datasets are available on the NCBI Sequence Read Archive BioProject-PRJNA837466 (https://www.ncbi.nlm.nih.gov/sra/PRJNA837466, accessed on 17 May 2022).

## 3. Results

### 3.1. Assessment of Primer Specificity and Amplification Efficiency

Eleven candidate reference genes were selected from among previously used genes for normalization in cucumber or other species. The sequences of the primers and their associated parameters are listed in Table 1. To identify the primer specificity, the amplified products were analyzed using 2% agarose gel electrophoresis. Only one clear band of the expected size appeared in each lane, and no primer dimers or non-specific amplification could be detected (Figure 2). The specific bands were cut off and sent to Sanger-sequencing. Blastn verified that the amplicons were from the targeted genes (data not shown).

To further assess the primer specificity, RT-qPCR was conducted. A single peak in the melting curve was obtained after the amplification of all 11 genes (Figure 3). The corresponding PCR amplification efficiencies (E) and the linear relationships between the Ct values and the log-transformed copies indicated by correlation coefficients (R^2^) for all the tested reference genes were calculated from the RT-qPCR data (Table 1). The E values ranged from 91.99% to 96.24%, which were all within the acceptable range of 90–105%. Furthermore, the R^2^ values ranging from 0.9924 to 0.9990 confirmed the specificity of the primer pairs.

### 3.2. Expression Levels and Variation in Candidate Reference Genes

In order to show the different transcriptional levels among the 11 candidate genes, the average Ct values were determined using all the experimental samples. The Ct values of the candidate genes in all the samples are illustrated as a box plot in Figure 4. The average Ct values of the 11 genes ranged from 18.272 to 26.678. The *GAPDH* gene showed the highest expressive abundance, with the lowest Ct value of 17.727, while the *F-BOX* showed the lowest abundance with the highest Ct value of 27.548 (Figure 4). The Ct values for *L23* (20.107–23.001) and *TUA* (20.796–25.152) showed the largest variation for one gene, whereas those for *CYP* (17.821–19.435) and *GAPDH* (17.727–19.489) showed the smallest variation.

To further evaluate the expression stability of the candidate reference genes, four methods based on different algorithms were used to calculate the expression stability: geNorm, NormFinder, ΔCt method, and BestKeeper.

### 3.3. Analyses of Candidate Reference Gene Stability Using Four Different Types of Software

#### 3.3.1. geNorm Analysis

The raw Ct values from RT-qPCR were transformed into quantities for the geNorm analysis. The average gene-expression stability measurement (M-value) of the 11 candidate genes were obtained through the program, and the genes were ranked based on the M-value, from highest to lowest. The genes with the highest M-values showed lower stability, while the genes with the lowest M-values indicated higher stability. All of the 11 selected genes showed an acceptable level of expression stability with M < 0.702, which was below the cut-off value of 1.5 suggested by geNorm (Figure 5a). *F-BOX* and *UBI-1* were the most stable genes, followed by *EF1α*, and the *TUA* and *TIP41* genes showed the least stability according to geNorm. In addition, the optimal number of reference genes required for reliable normalization was also determined by geNorm. The pairwise value for two genes (V_2/3_) was 0.107, which was lower than the threshold of 0.15 (Figure 5b). This result indicated that two reference genes were sufficient for accurate analyses of the gene expression in the context of PNRSV infection.

#### 3.3.2. NormFinder Analysis

NormFinder was used to evaluate the stability of the reference genes based on the SV parameter. Lower SV values meant higher stability. The SVs of the 11 candidate genes are shown in Table 2 and ranked in Figure 5c. The results of the NormFinder analysis illustrated that the two most stable genes were *EF1α* and *F-BOX*, and the two least stable candidate genes were *TUA* and *TIP41*.

#### 3.3.3. Δ.Ct Analysis

The comparative ΔCt method identified that *EF1α* and *F-BOX* were the most two suitable genes, and *TUA* and *TIP41* were the least suitable (Figure 5d). The ranking order of the 11 candidate genes was similar to that obtained by NormFinder analysis.

#### 3.3.4. Bestkeeper Analysis

The three variables, SD, coefficient of correlation (r), and CV play key roles in gene-expression stability analyses using the Bestkeeper program. Candidate genes with SD values > 1 should be excluded, since they are considered inconsistent. In the study, the analysis of the data from all the candidate genes using Bestkeeper showed that all the candidate reference genes had SD values < 1. Across all the candidate genes, *TUA* had relatively high SD and CV values (SD *_TUA_* = 0.81 and CV *_TUA_* = 3.66), which can be considered to have been eliminated in the subsequent analysis (Table 3). In addition, the Bestkeeper analysis showed that the *p*-values of the genes *CYP*, *GAPDH*, *UBI-ep*, *CACS*, and *TIP41* were over 0.05 (0.105 < *p*-value < 0.216), which meant they could also be excluded from the subsequent ranking. The remaining candidate genes were ranked from the most stable, with the highest coefficient of correlation, to the least stable, with the lowest value. *EF1α* (r = 0.873; *p*-value = 0.001) was the most stable gene, followed by *PP2A* (r = 0.845; *p*-value = 0.001) and *F-BOX* (r = 0.826; *p*-value = 0.001) (Table 2). These three genes (*EF1α*, *F-BOX*, and *PP2A*) were also the most stable genes identified by NormFinder analysis and the ΔCt method.

#### 3.3.5. RefFinder Analysis

To obtain a final overall ranking, RefFinder, a web-based comprehensive evaluation platform, was performed, based on the results of the above four programs. According to the RefFinder analysis, *EF1α* and *F-BOX* were the top two suitable reference genes for normalizing the transcripts in PNRSV-infected *C. sativus*, whereas *TIP41* and *TUA* were the least stable genes (Table 2). According to the optimal number of reference genes evaluated by geNorm, *EF1α* and *F-BOX* were selected as the optimal multiple reference genes for normalization in PNRSV-infected *C. sativus* at 12 dpi.

### 3.4. Influence of Different Selection of Reference Genes on the Relative Expression of Target Genes

To investigate the influences of different reference genes on the expression of the target mRNA, we selected *AGO1*, *AGO4*, and *RDR6*, which are the important genes in RNA interference (RNAi)-based antiviral immunity (Figure 6), for additional RT-qPCR analyses. To identify the primer specificity, the amplified products were analyzed using 2% agarose gel electrophoresis and the melting curve (Appendix A). The most stable genes, *EF1α* and *F-BOX*, were used in combination for normalization, while the less stable genes, *TIP41* and *TUA*, were also used as comparison. The relative expression levels of *AGO1* and *AGO4* showed significant upregulation, and the relative expression level of *RDR6* showed no significant difference after PNRSV infection when we normalized using *EF1α* and *F-BOX* in combination. The *AGO1*, *AGO4*, and *RDR6* genes exhibited similar expression trends when using *TIP41* as the reference gene, while the upregulated folds were much larger than when using *EF1α* and *F-BOX* in combination, which indicated that the relative expressions were obviously overestimated (Figure 6a–c). By contrast, the relative expression of *AGO1* and *AGO4* in the PNRSV-inoculated *C. sativus* showed quite different trends when *TUA* was used for normalization, and did not show any differences compared with mock-inoculated plants.

Moreover, the relative expression levels of *AGO1*, *AGO4*, and *RDR6* were calculated by high-throughput RNA sequencing data, which indicated that the expression of the *AGO1* in the PNRSV-inoculated *C. sativus* was 2.28 times that in healthy plants, the expression of *AGO4* in PNRSV-inoculated *C. sativus* was 1.61 times that in healthy plants, and the expression level of *RDR6* showed no significant difference after PNRSV infection (Figure 6d). The relative expression levels of *AGO1*, *AGO4*, and *RDR6* by transcriptome analyses were similar to the RT-qPCR results, which validated the use of EF1α and F-BOX as normalization genes.

## 4. Discussion

RT-qPCR is one of the most widely used methods for quantifying gene expression changes [5]. Relative changes in gene expression can be determined without knowing the absolute quantity of the reference genes, while the accuracy and repeatability of RT-qPCR analysis highly depends on many factors, such as sample quality, RNA quality and integrity, primer specificity, PCR conditions and amplification efficiency, etc. Based on these factors, the selection of a suitable reference gene is of primary importance [18,47]. In this study, we proposed a systematic process through which to identify the optimal reference genes in the leaf tissues of *C. sativus* under PNRSV infection. Firstly, the primer specificity of the candidate reference genes was experimentally validated by PCR amplification, Sanger sequencing, and melting curves. Secondly, the Ct value, E value, and R^2^ were achieved for the tested reference genes. The suitable Ct values, 95% < E < 105%, and R^2^ > 99%, are the prerequisite for using the 2^-ΔΔCt^ method for data analysis [48]. Subsequently, four methods based on different algorithms and one platform for ranking order were used to evaluate the expression stability of the candidate reference genes [48,49]. geNorm is considered one of the best methods to determine the most stable genes and the optimal number of genes [16]. *F-BOX*, *UBI-1*, and *EF1α* were the most stable genes and *TUA* and *TIP41* showed the least stability according to geNorm. At the same time, geNorm indicated that two reference genes were sufficient in our search. NormFinder uses the variation among candidate genes to rank, taking into account intra-group and inter-group variation, and reduces bias through co-regulation [17]. When we used NormFinder, *EF1α* and *F-BOX* were the most stable genes. The ranking results showed almost no differences between the NormFinder and ΔCt methods. BestKeeper is a useful approach for calculating the coefficients of correlation, SD and CV [18]. The *EF1α* was the most stable gene and *TUA* had a high r value, but its high *SD* and *CV* values suggested instability. Combining these four methods, *EF1α* and *F-BOX* were the top-ranked reference genes, while *TUA* and *TIP41* were the lowest=ranked, according to the RefFinder analysis. Finally, three virus-immunity-associated genes were selected to validate the importance of the selection of suitable reference genes.

Several viruses infecting woody plants in the genus *Ilarvirus* cause serious damages in the fruit industry [39]. PNRSV induced significant losses in stone-fruit trees when it broke out in Turkey, India, etc. [50,51]. The ApMV and ApNMV viruses were identified as having induced apple mosaic disease, which is characterized by the symptoms of mosaic leaves and occurs widely in major apple-producing areas around the world [52,53]. PDV, another ilarvirus infecting stone-fruit species, caused deteriorated fruit marketability and reduced fruit yields [39]. The fact that the infectious clones of the above four viruses caused clear symptoms in cucumbers, the breakthrough of transgenic technology, and the establishment of VIGS systems on cucumbers, have prompted the application of cucumber as a useful and crucial experimental material to facilitate studies on the infection mechanism of ilarviruses. The study of pathogenic mechanisms is very important in the unravelling the viral infections and provide clues for the development of resistant materials [54]. Therefore, the validation of reference genes for gene expression studies using RT-qPCR after viral infection in *C. sativus* is highly necessary. Recently, some studies have identified a variety of reference genes in cucumbers at different developmental stages and under different biotic or abiotic stresses. *TUA* and *UBI-ep* were the most stably expressed genes when the cucumber seedlings at the second true leaf stage were treated with hormones, and *EF1α* showed a relatively stable expression level when the seedlings were treated with salt and drought stress [47,55]. The genes *CACS*, *TIP41*, *F-BOX*, and *EF1α* showed the highest expression stability under different nitrogen nutrition regimens and the combined use of three or four references helped to obtain reliable results [48]. In terms of biotic stress, *EF1α* and *GAPDH* were the most reliable reference genes for normalizing the miRNA expression after cucumber green mottle mosaic virus infection in leaf, root, and stem samples [56]. In our research, we identified *EF1α* and *F-BOX* as the most suitable reference genes among the eleven candidate genes for normalization in the PNRSV-infected leaf tissues of *C. sativus* at 12 dpi. By contrast, the validated reference genes of *TUA*, *TIP41*, *CACS*, and *UBI-ep* in the previous experiments were relatively unstable and ranked lower in the stability ranking, which further indicated that there is no universal reference gene that can be stably expressed and used under all conditions for all kinds of tissue. The stability of reference genes depends on the experimental parameters, and the selection of reference genes should be studied in advance, case by case, to prevent subsequent false data interpretation and conclusions.

RNAi-mediated plant immunity is believed to be a universal pathway against viruses [57]. To further validate our conclusion, we investigated the relative expression of *AGO1*, *AGO4*, and *RDR6* using the normalization factor generated with *EF1α* and *F-BOX* in combination, which showed that *AGO1* expression was greatly and significantly upregulated in PNRSV-infected leaf tissues compared with the mock-inoculated control plants, and suggested that *AGO1* may play an important role in PNRSV infection. Furthermore, when using *TUA* or *TIP41* as reference genes, the differential expression of *AGO1* and *AGO4* was ignored or overemphasized. To support these conclusions, transcriptome-wide high-throughput RNA sequencing was conducted to provide a more accurate, unbiased estimate of the transcript abundance of *AGO1*, *AGO4*, and *RDR6*. The changes in the expression trends were similar to the RT-qPCR results. These findings further indicate that the use of appropriate reference genes is crucial to obtain accurate results.

In this study, we assessed eleven candidate reference genes in *C. sativus* in the context of PNRSV infection and determined that the combined use of *EF1α* and *F-BOX* is the best choice. This study represents the first attempt to select suitable reference genes in *C. sativus* in the context of ilarvirus infection, which will facilitate the functional study of genes related to viral infection.

## Figures and Tables

**Figure 1 viruses-14-01269-f001:**
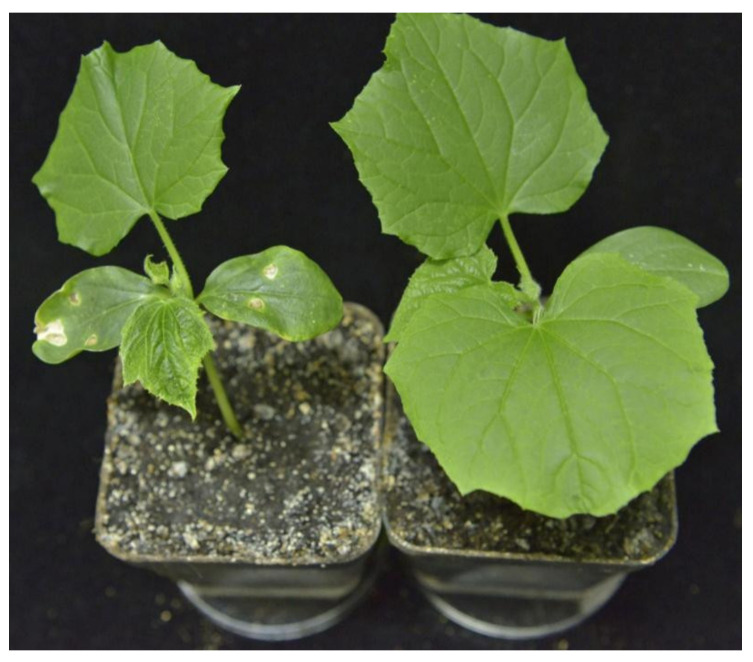
The symptoms of prunus necrotic ringspot virus (PNRSV)-infected *Cucumis sativus* (**left**) and healthy plant (**right**).

**Figure 2 viruses-14-01269-f002:**
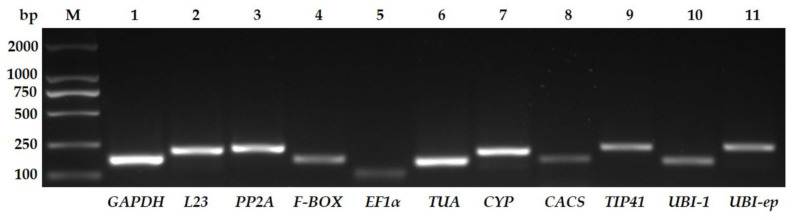
Amplification products of the eleven candidate reference genes. Lane M, Trans2K DNA marker; lane 1, *GAPDH*; lane 2, *L23*; lane 3, *PP2A*; lane 4, *F-BOX*; lane 5, *EF1α*; lane 6, *TUA*; lane 7, *CYP*; lane 8, *CACS*; lane 9, *TIP41*; lane 10, *UBI-1*; lane 11, *UBI-ep*.

**Figure 3 viruses-14-01269-f003:**
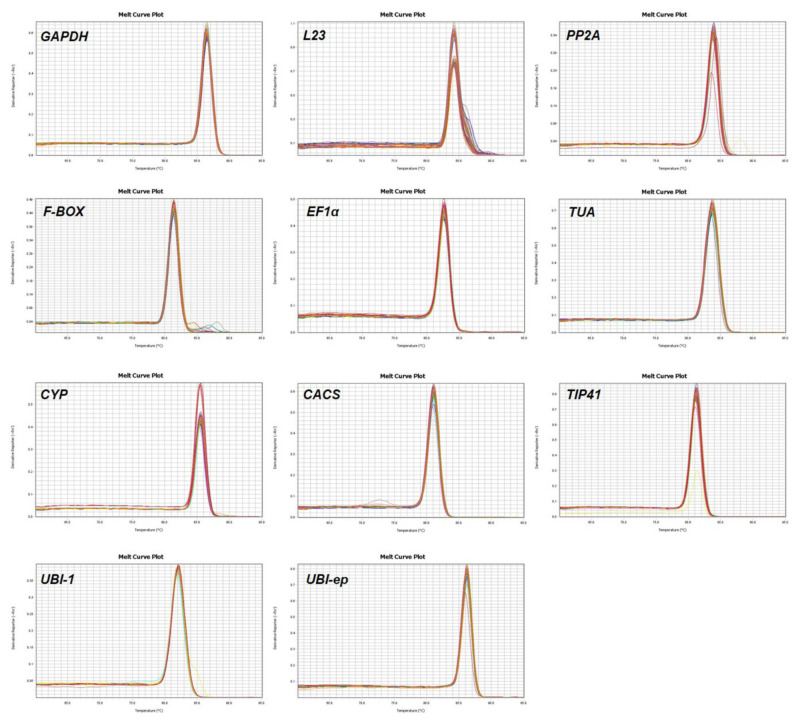
Dissociation curves of the eleven reference genes evaluated under experimental conditions, each showing a single peak.

**Figure 4 viruses-14-01269-f004:**
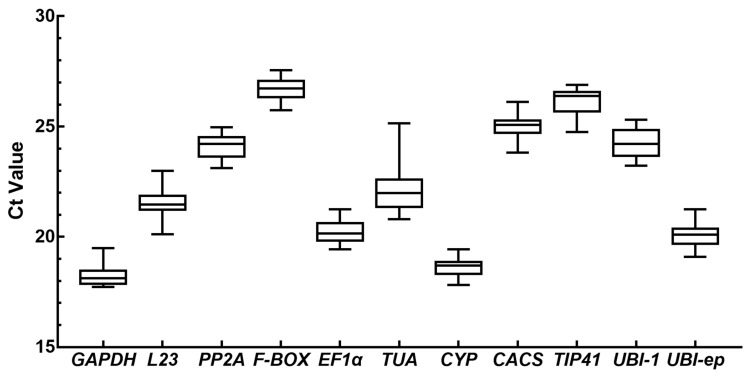
Expression levels of the eleven candidate reference genes evaluated in prunus necrotic ringspot virus (PNRSV)-infected *C. sativus*. Values are given as the cycle threshold (Ct, mean of triplicate samples) and are inversely proportional to the amount of template. The three lines of the box represent the 25th quartiles, median, and 75th quartiles. The whiskers represent minimum and maximum values.

**Figure 5 viruses-14-01269-f005:**
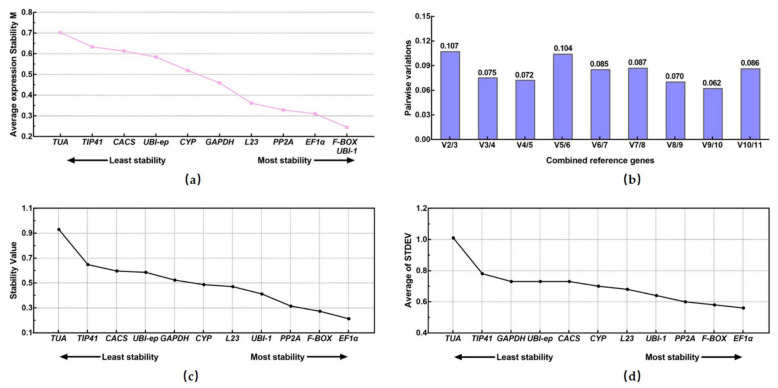
Values of gene-expression stability of the eleven candidate genes. (**a**) geNorm-expression stability measurement (M-value). A lower M-value indicates more stable gene expression. (**b**) The optimal number of reference genes required for effective qRT-PCR data normalization by geNorm. Pairwise variation (V_n_/V_n+1_) between the normalization factors NF_n_ and NF_n+1_ (NF_n_ is the expression values for the *n* first-ranked candidate reference genes). (**c**) Expression stability of the candidate reference genes analyzed by NormFinder. A lower stability value indicates more stable gene expression. (**d**) The STDEV values analyzed by ∆Ct method. A lower value indicates more stable gene expression.

**Figure 6 viruses-14-01269-f006:**
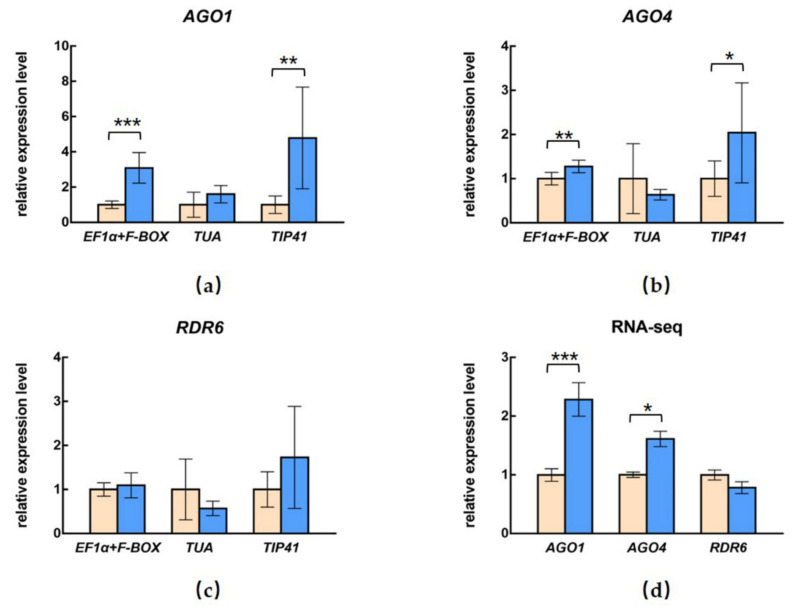
Relative expression of (**a**) *AGO1*, (**b**) *AGO4*, and (**c**) *RDR6* in PNRSV-infected *C. sativus* with different reference genes. *EF1α* and *F-BOX* were the most stable genes, while *TIP41* and *TUA* were the least stable genes. (**d**) The relative expression levels of *AGO1*, *AGO4*, and *RDR6* were calculated by high-throughput RNA sequencing data. * means *p* < 0.05, ** means *p* < 0.01, and *** means *p* < 0.0001 (based on t-test analysis).

**Table 1 viruses-14-01269-t001:** Primers used in this study. Tm, annealing temperature; E: PCR amplification efficiency; R^2^: correlation coefficient.

Gene	Full Name	Accession Number	Sequence (5′-3′)	Amplicon Length (bp)	Tm (°C)	E (%)	R^2^	References
*GAPDH*	*Glyceraldehyde-3-phosphate dehydrogenase*	NM_001305758	F: CATCAAGGGAACAATGACTACCAC R: TAAGAGAAGGGAGGACCAAGGC	161	58	95.21	0.9928	This study
*L23*	*60S ribosomal protein*	XM_004140527	F: AACAAGTTTCGGATGTCACTGG R: CAATAACAGCAGGCAAGACCTTC	211	58	93.92	0.9946	This study
*PP2A*	*Protein phosphatase 2A*	XM_004146712	F: TGGAAGCACTTCACCGACCTCT R: CCAGCACCACGCGGAGATATTC	224	58	94.37	0.9992	This study
*F-BOX*	*F-box protein*	XM_031885920	F: AAGCGTCTATTCCTTGTTGGTAGG R: CAAAGCACTCAAATCTCTCCGC	166	58	96.24	0.9925	This study
*EF1α*	*Elongation factor 1-α*	DQ341381	F: ATTCAAGTATGCCTGGGTGC R: CAGTCAGCCTGTGATGTTCC	174	58	92.10	0.9958	Wan et al., 2012 [46]
*TUA*	*α-Tubulin*	AJ715498	F: CATTCTCTCTTGGAACACACTGA R: TCAAACTGGCAGTTAAAGATGAAA	106	58	95.74	0.9938	Wan et al., 2010 [47]
*CYP*	*Cyclophilin*	AY942800	F: GGAAATGGTACAGGAGGTG R: CATACCCTCAACGACTTGAC	88	58	95.02	0.9984	Wan et al., 2010 [47]
*CACS*	*AP-2 complex subunit mu-1*	GT035008	F: GTGCTTTCTTTCTGGAATGC R: TGAACCTCGTCAAATTTACACA	158	58	93.12	0.9924	Warzybok et al., 2013 [48]
*TIP41*	*TIP41-like family protein*	GW881871	F: CAACAGGTGATATTGGATTATGATTAC R: GCCAGCTCATCCTCATATAAG	221	58	91.99	0.9928	Warzybok et al., 2013 [48]
*UBI-1*	*Ubiquitin-like protein*	AF104391	F: CCAAAGCACAAGCAAGAGAC R: AGTAGGTTGTCTTATGGCGC	143	58	93.29	0.9990	Wan et al., 2010 [47]
*UBI-ep*	*Ubiquitin extension protein*	AY372537	F: CACCAAGCCCAAGAAGATC R: TAAACCTAATCACCACCAGC	220	58	92.93	0.9955	Wan et al., 2010 [47]
*AGO1*	*Argonaute 1*	XM_011651337	F: CGTTTCCCAGTGCTGTTTGAC R: GCCAATCTTGAGAAGCCACAAC	245	58	94.79	0.9987	This study
*AGO4*	*Argonaute 4*	XM_011655229	F: GTCAAGCATCGGAACATCG R: TCCTTCTCAACGAATCGGAC	223	58	94.73	0.9982	This study
*RDR6*	*RNA dependent RNA polymerase 6*	XM_011650266	F: TGTCTCACCTGACTCTGCTGCT R: ATTCCACTCGAAGGCCCTCTCC	228	58	95.93	0.9961	This study

**Table 2 viruses-14-01269-t002:** Expression stability ranking of the nine candidate reference genes calculated by four different types of software. M-value: stability measurement; r: coefficient of correlation.

Gene Name	geNorm		NormFinder		Delt Ct		BestKeeper		RefFinder	
	M-Value	Ranking	Stability	Ranking	Stability	Ranking	r Value	Ranking	Stability	Ranking
*EF1α*	0.310	3	0.213	1	0.56	1	0.873	1	1.73	1
*F-BOX*	0.244	1	0.273	2	0.58	2	0.826	4	2.00	2
*UBI-1*	0.244	1	0.412	4	0.64	4	0.765	5	3.46	3
*CYP*	0.519	7	0.487	6	0.70	6	0.327	7	3.98	4
*PP2A*	0.328	4	0.315	3	0.60	3	0.845	3	4.12	5
*GAPDH*	0.459	6	0.523	7	0.73	7	0.387	8	4.92	6
*L23*	0.361	5	0.471	5	0.68	5	0.761	6	5.44	7
*UBI-ep*	0.584	8	0.586	8	0.73	7	0.356	9	7.11	8
*CACS*	0.613	9	0.596	9	0.73	7	0.413	10	8.13	9
*TIP41*	0.633	10	0.648	10	0.78	10	0.421	11	10.00	10
*TUA*	0.702	11	0.930	11	1.01	11	0.856	2	11.00	11

**Table 3 viruses-14-01269-t003:** Descriptive statistics of the eleven candidate reference genes by Bestkeeper analysis. *n*: number of samples; Geo Mean (Ct): geometric mean of Ct value; AR mean (Ct): arithmetic mean of Ct value; Min and Max (Ct): extreme values of Ct; SD: standard deviation; CV: coefficient of variation; Min and Max (x-fold): extreme values of expression levels; SD (±x-fold): standard deviation of the absolute regulation coefficients; r: coefficient of correlation.

	Reference Gene										
	*EF1α*	*TUA*	*PP2A*	*F-BOX*	*UBI-1*	*L23*	*CYP*	*GAPDH*	*UBI-ep*	*CACS*	*TIP41*
Ranking	1	2	3	4	5	6	7	8	9	10	11
*n*	16	16	16	16	16	16	16	16	16	16	16
Geo Mean (Ct)	20.21	22.19	24.13	26.67	24.24	21.48	18.62	18.27	20.04	24.99	26.10
AR Mean (Ct)	20.22	22.22	24.13	26.68	24.25	21.49	18.62	18.27	20.05	24.99	26.11
Min (Ct)	19.43	20.80	23.11	25.74	23.22	20.11	17.82	17.73	19.09	23.82	24.75
Max (Ct)	21.24	25.15	24.97	27.55	25.31	23.00	19.44	19.49	21.25	26.12	26.89
**SD (±Ct)**	**0.40**	**0.81**	**0.48**	**0.41**	**0.53**	**0.47**	**0.31**	**0.36**	**0.41**	**0.43**	**0.54**
**CV (%Ct)**	**1.99**	**3.66**	**1.97**	**1.54**	**2.17**	**2.18**	**1.65**	**1.98**	**2.07**	**1.71**	**2.09**
Min (x-fold)	−1.72	−2.63	−2.02	−1.91	−2.03	−2.59	−1.74	−1.45	−1.94	−2.24	−2.55
Max (x-fold)	2.04	7.79	1.79	1.83	2.09	2.87	1.76	2.33	2.31	2.19	1.73
SD (±x-fold)	1.32	1.76	1.39	1.33	1.44	1.38	1.24	1.29	1.33	1.35	1.46
**coeff. of corr. (r)**	**0.873**	**0.856**	**0.845**	**0.826**	**0.765**	**0.761**	**0.327**	**0.387**	**0.356**	**0.413**	**0.421**
*p*-value	0.001	0.001	0.001	0.001	0.001	0.001	0.216	0.138	0.176	0.112	0.105

## Data Availability

Not applicable.

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
