# Peer review of "Selection and Validation of Reference Genes for Gene Expression Studies Using Quantitative Real-Time PCR in Prunus Necrotic Ringspot Virus-Infected *Cucumis sativus"

_viruses, 2022, doi:10.3390/v14061269_

Round 1
Reviewer 1 Report
Dear Editor, I am sending you my review of the manuscript entitled
“Selection and Validation of Reference Genes for Gene Expression Studies Using Quantitative Real-time PCR in Prunus Necrotic Ringspot Virus-infected Cucumis Sativus”, by Dong et al. The manuscript evaluates the stability of eleven reference genes using different methods for doing so. The topic of the manuscript is interesting and the information generated can be useful for other researchers. However, my main objection is with the justification of the work. The authors justified its importance in the use of cucumber plants as a model plant in the research of illarvirus from woody plants, instead of the importance of PNRSV in cucumber crop. More information about this should be included in the manuscript. Also the discussion should be deeper for a journal such as Viruses.
I also have some comments and suggestions:
Material and methods: mechanical inoculation should be enough to induce Illarvirus symptoms in cucumber plants. Some Figures showing symptoms have to be added in the manuscript.
During the manuscript, we can read Cucumis sativus and Cucumis Sativus, correct it (first option). Also, the complete name should be used only the first time, the thereafter C. sativus is enough.
The authors use the gen name EF1α in the text, and EF1A in the graphics. Is it the same gene?
Therefore, in my opinion, after all these changes the manuscript should be accepted in Viruses.
Author Response
Response to Reviewer 1 Comments
Point: my main objection is with the justification of the work. The authors justified its importance in the use of cucumber plants as a model plant in the research of illarvirus from woody plants, instead of the importance of PNRSV in cucumber crop. More information about this should be included in the manuscript. Also the discussion should be deeper for a journal such as Viruses.
Response: We added the severe symptom of PNRSV-infected cucumber plant, which show the importance of PNRSV in cucumber plant (Figure 1). We also enriched the discussion part (lines 327-338, lines 368-372, lines 380-383).
Point: Material and methods: mechanical inoculation should be enough to induce Illarvirus symptoms in cucumber plants. Some Figures showing symptoms have to be added in the manuscript.
Response: The method of agroinfiltration was used to inoculate cucumber plants, which contribute to subsequent quantitative analysis. We added ‘Figure 1’ as suggested.
Point: During the manuscript, we can read Cucumis sativus and Cucumis Sativus, correct it (first option). Also, the complete name should be used only the first time, the thereafter C. sativus is enough.
Response: We revised as suggested throughout the manuscript.
Point: The authors use the gen name EF1α in the text, and EF1A in the graphics. Is it the same gene?
Response: We used the gene name EF1α in the text and changed the EF1A in the graphics (Figure 4 and Figure 5) to EF1α.
Reviewer 2 Report
Dong et al. investigated transcript level changes of cucumber plants infected with Prunus necrotic ringspot virus. Their manuscript presents a list of genes for normalization of real-time quantitative PCR assays suitable for their experimental system, and they recommend the combined use of EF1α and F-BOX as normalization genes.
The study is interesting, describes useful resources for plant virologists, and it is within the journal scope.
There are however major concerns that should be addressed:
- Figure legends should be revised to provide complete information.
- Many viruses infect cucumber. It would be interesting to validate the stability of the proposed normalization genes in cucumber samples infected with other viruses.
- Hormone signaling pathways have large effects on plant transcriptome and antiviral immunity (Alazem et al. 2019 https://doi.org/10.3390/ijms20102538 ). It would be interesting to validate the stability of the proposed normalization genes in cucumber samples treated with major hormones.
- Fig. 5 nicely shows that use of different normalization genes can lead to very contrasting results. I would recommend the authors to support their conclusions using an alternative technique. For instance, transcriptome-wide microarray or RNA-seq analyses might provide a more accurate, unbiased estimate of transcript abundance than RT-qPCR (e.g. Pasin et al. 2020 https://doi.org/10.1016/j.xplc.2020.100099 ) and could be used to validate EF1α and F-BOX as normalization genes.
Author Response
Point: Figure legends should be revised to provide complete information.
Response: We enriched the figure legends as suggested (lines 237-241 and lines 310-312).
Point: Many viruses infect cucumber. It would be interesting to validate the stability of the proposed normalization genes in cucumber samples infected with other viruses. Hormone signaling pathways have large effects on plant transcriptome and antiviral immunity (Alazem et al. 2019 https://doi.org/10.3390/ijms20102538 ). It would be interesting to validate the stability of the proposed normalization genes in cucumber samples treated with major hormones.
Response: It is critically important to evaluate the stability of the candidate reference genes experimentally for accurate quantification analyses case-by-case, which varies with the conditions, such as specific species, the sample or tissue types, different treatments, experimental conditions, etc. The previous research showed that TUA and UBI-ep were the most stably expressed genes when the cucumber seedlings were treated with hormones (Wan et al., 2010, doi:10.1016/j.ab.2009.12.008.). We would evaluate the stability of the candidate reference genes in cucumber samples infected with other viruses or treated with different hormones, maybe in the subsequent research.
Point: Fig. 5 nicely shows that use of different normalization genes can lead to very contrasting results. I would recommend the authors to support their conclusions using an alternative technique. For instance, transcriptome-wide microarray or RNA-seq analyses might provide a more accurate, unbiased estimate of transcript abundance than RT-qPCR (e.g. Pasin et al. 2020 https://doi.org/10.1016/j.xplc.2020.100099 ) and could be used to validate EF1α and F-BOX as normalization genes.
Response: We added the results of the relative expression changes of AGO1, AGO4 and RDR6 by the previous conducted high-throughput RNA-sequencing analyses (Lines 301-307)
Round 2
Reviewer 2 Report
The authors have nicely enhanced the previous manuscript version by including comparisons of RT-qPCR quantification and high-throughput RNA-sequencing analyses.
I have some minor comments:
- Materials and methods need to be updated to describe high-throughput RNA-sequencing and data analysis.
- Fig. 6, it would be nice to include for each gene (AGO1, AGO4 and RDR6) additional plot bars or panels to present RNA-seq quantification. This would make much easier to compare RT-qPCR and RNA-seq results.
- I would recommend to make high-throughput RNA-sequencing data available from public repositories (e.g. at NCBI SRA www.ncbi.nlm.nih.gov/sra)
Author Response
Response to Reviewer 2 Comments-2nd
Point: Materials and methods need to be updated to describe high-throughput RNA-sequencing and data analysis.
Response: We enriched high-throughput RNA-sequencing and data analysis in materials and methods in lines 179 to 196.
Point: Fig. 6, it would be nice to include for each gene (AGO1, AGO4 and RDR6) additional plot bars or panels to present RNA-seq quantification. This would make much easier to compare RT-qPCR and RNA-seq results.
Response: We added figure as suggested in Figure 6.
Point: I would recommend to make high-throughput RNA-sequencing data available from public repositories (e.g. at NCBI SRA www.ncbi.nlm.nih.gov/sra)
Response: We submit the RNA-sequencing data (SRA BioProject-PRJNA837466) to NCBI SRA.

This manuscript is a resubmission of an earlier submission. The following is a list of the peer review reports and author responses from that submission.